# Multidisciplinary Team-Based Palliative Care for Heart Failure and Food Intake at the End of Life

**DOI:** 10.3390/nu13072387

**Published:** 2021-07-13

**Authors:** Tatsuhiro Shibata, Kazutoshi Mawatari, Naoko Nakashima, Koutatsu Shimozono, Kouko Ushijima, Yumiko Yamaji, Kumi Tetsuka, Miki Murakami, Kouta Okabe, Toshiyuki Yanai, Shoichiro Nohara, Jinya Takahashi, Hiroki Aoki, Hideo Yasukawa, Yoshihiro Fukumoto

**Affiliations:** 1Division of Cardiovascular Medicine, Department of Internal Medicine, Kurume University School of Medicine, Kurume 830-0011, Japan; shibata_tatsuhiro@med.kurume-u.ac.jp (T.S.); mawatari_kazutoshi@kurume-u.ac.jp (K.M.); shimozono_koutatsu@med.kurume-u.ac.jp (K.S.); okabe_kouta@med.kurume-u.ac.jp (K.O.); yanai_toshiyuki@med.kurume-u.ac.jp (T.Y.); nohara_shoichiro@med.kurume-u.ac.jp (S.N.); takahashi_jinya@med.kurume-u.ac.jp (J.T.); yahideo@med.kurume-u.ac.jp (H.Y.); 2Kurume University Hospital Palliative Care Team, Kurume University, Kurume 830-0011, Japan; nakashima_naoko@kurume-u.ac.jp (N.N.); yoshigai_miki@kurume-u.ac.jp (M.M.); 3Department of Nursing, Kurume University Hospital, Kurume 830-0011, Japan; ushijima_kouko@kurume-u.ac.jp (K.U.); yumi0802minami@yahoo.co.jp (Y.Y.); tetsuka_kumi@kurume-u.ac.jp (K.T.); 4Cardiovascular Research Institute, Kurume University, Kurume 830-0111, Japan; haoki@med.kurume-u.ac.jp

**Keywords:** heart failure, palliative care, end-of-life care discussion, advance care planning, food intake, artificial nutrition

## Abstract

Traditionally, patients with end-stage heart failure (HF) have rarely been involved in end-of-life care (EOLC) discussions in Japan. The purpose of this study was to examine the impact of HF-specific palliative care team (HF-PCT) activities on EOLC discussions with patients, HF therapy and care, and food intake at the end of life. We retrospectively analyzed 52 consecutive patients with HF (mean age, 70 ± 15 years; 42% female) who died at our hospital between May 2013 and July 2020 and divided them into two groups: before (Era 1, n = 19) and after (Era 2, n = 33) the initiation of HF-PCT activities in June 2015. Compared to Era 1, Era 2 showed a decrease in invasive procedures, an increase in opioid and non-intubating sedative use for symptom relief, improved quality of meals at the end of life, and an increase in participation in EOLC discussions. The administration of artificial nutrition in the final three days was associated with non-ischemic cardiomyopathy etiology, the number of previous hospitalizations for HF, and multidisciplinary EOLC discussion support. HF-PCT activities may provide an opportunity to discuss EOLC with patients, reduce the burden of physical and psychological symptoms, and shift the goals of end-of-life nutritional intake to ensure comfort and quality of life.

## 1. Introduction

Recent developments in new drugs, monitoring systems, and device therapies have evolved heart failure (HF) therapy; however, these developments may stabilize HF but rarely cure it. Furthermore, these advances are often only available for a limited number of patients. During HF progression, patients experience a high symptom burden and poor quality of life similar to that reported by patients with cancer [1,2]. HF often follows an unpredictable illness trajectory, with stable periods interrupted by exacerbations and sometimes resulting in sudden cardiac death, leading to difficulties in estimating survival [3]. When focusing on end-of-life decision making, such prognostic uncertainty complicates the patients’ plans concerning their end-of-life wishes and sometimes leads to an overestimation of survival [4]. Furthermore, traditionally in Japan, patients rarely participate in discussions about their own goals and preferences for end-of-life care (EOLC), because their families may hesitate to bring bad news. As patients tend to lose their decision-making ability toward the end of life [5], they sometimes have no opportunity to express their wishes and preferences regarding their EOLC. Until recently in Japan, these situations often led to providing life-prolonging treatment to critically ill patients, regardless of the medical futility [6]. In addition, at the end-stage of HF, associated symptoms such as anorexia, nausea, vomiting, and dyspnea are observed, and food intake decreases [7]. In patients with end-stage HF, the goal of nutritional care is to optimize quality of life and comfort. Even though patients eventually lose their appetite spontaneously, artificial nutrition has often been attempted in the past.

Palliative care comprises a multidisciplinary team approach for patients and their families facing serious illnesses that focuses on improving quality of life and death. Recently, palliative care has been recommended by the Japanese Circulation Society (JCS)/Japanese Heart Failure Society (JHFS) and American College of Cardiology (ACC)/American Heart Association (AHA) HF guidelines [8,9]. The core elements of a multidisciplinary palliative care team (PCT) include not only expert assessment of physical and psychosocial distress but also the establishment of care goals and support for advance care planning (ACP) and complex decision-making, including EOLC discussion. In Japan, a multidisciplinary PCT is available in most of the regional cancer centers; however, it is only available in 9% of JCS-authorized cardiology training hospitals [10]. One of the major reasons is that PCT intervention had not been reimbursed for patients with HF by April 2018. Given that these services are not yet widely available, there has been insufficient clinical data regarding PCTs in patients with HF in Japan. Moreover, even in Western countries, only limited evidence is available regarding the patient-centered EOLC discussions of inpatient PCTs with patients with HF [11,12].

Therefore, the main purpose of this study was to compare changes in HF therapies in terms of palliative care and EOLC discussions, with a focus on food intake at the end of life in HF, before and after the initiation of HF-specific PCT (HF-PCT) activities at our institute.

## 2. Materials and Methods

### 2.1. Study Design and Population

The study was performed at Kurume University Hospital, a 1000-bed tertiary medical center located in the southern part of Fukuoka Prefecture, Japan. We retrospectively analyzed the medical records of 215 consecutive patients who died at the Division of Cardiology of our hospital between May 2013 and July 2020. Among these, we excluded 163 cases of non-HF deaths, including acute myocardial infarction, and patients treated only in the intensive care unit. HF was diagnosed by at least two specialist cardiologists based on the Framingham criteria [13]. Thus, we conducted a final analysis of 52 patients with HF who died in our department. These patients were divided into two groups according to their time of death before (Era 1; May 2013 to May 2015) and after (Era 2; June 2015 to July 2020) the HF-PCT activity started in June 2015. In this study, a multidisciplinary palliative approach for patients and their families was provided by an HF-PCT, which consisted of cardiologists, palliative care physicians, nurses (inpatient, outpatient, and a certified palliative care nurse), pharmacists, a psychologist, a medical social worker, and a managerial dietitian. The HF-PCT was available for all HF patients and assisted the treatment of patients with HF through refractory symptom relief, the establishment of care goals, psychosocial support, support of ACP and EOLC discussions, and provision of nutritional support for EOLC.

The demographic and clinical information of each patient was extracted from the electronic medical records of Kurume University Hospital. We obtained data on the patients’ background, etiology of HF, duration of HF, comorbidities, echocardiographic findings, HF treatments, sedative medications, laboratory data, location of death, invasive procedures undergone before death (cardiopulmonary resuscitation, intubation, direct current shocks, and mechanical circulatory support) and length of hospital stay. We also assessed differences in the palliative care intervention, such as the use of opioids and sedative medications for refractory symptoms, psychiatric support, and multidisciplinary support for EOLC discussions, between the patients who died in Era 1 and those who died in Era 2. To assess the nutritional status of patients with end-stage HF, we examined the method of receiving nutrition three days prior to death and nutritional interventions (discontinuation of salt reduction, allowing non-hospital meals, and use of oral nutritional supplements) for patients who were maintaining oral intake at the time. We also examined factors associated with the use of artificial nutrition (total parenteral nutrition and tube feeding) three days prior to death.

### 2.2. Statistical Analysis

Continuous variables are presented as mean ± standard deviation (SD) or median (interquartile range (IQR)), as appropriate; they were compared using Student’s *t*-test. Categorical baseline variables are presented as numbers (percentage) and compared using the chi-square or Fisher’s exact test. Univariate associations between baseline characteristics and multidisciplinary support for end-of-life discussions were performed using univariate logistic regression. Variables relevant to the model were selected based on a univariate threshold *p*-value (≤0.05) and included in a multivariate logistic model to predict the odds of receiving artificial nutrition in the final three days prior to death. Adjusted odds ratios (ORs) and 95% confidence intervals (CIs) were calculated. Statistical significance was set at *p* < 0.05. All statistical analyses were performed using EZR (Saitama Medical Center, Jichi Medical University, Saitama, Japan), which is a graphical user interface for R (The R Foundation for Statistical Computing, Vienna, Austria). More precisely, it is a modified version of R Commander designed to add statistical functions that are frequently used in biostatistics.

## 3. Results

### 3.1. Patient Characteristics

Patient characteristics are shown in Table 1. Among the 52 patients, 22 (42%) were female, with an age of 70.0 ± 15.1 years. The median number of HF hospitalizations prior to death was 3 (IQR 1–5, and 42% of patients had non-ischemic cardiomyopathy (NICM) etiology. The median left ventricular ejection fraction was 35% (IQR 20–56%), and the median N-terminal B-type natriuretic peptide (NT-proBNP) was 12,683 pg/mL (IQR 5181–31,264 pg/mL). Nineteen patients who died during Era 1 and 33 patients who died during Era 2 were included in the analyses. There were no significant differences in demographics or clinical characteristics between the two groups, except for the etiology of “Others”.

### 3.2. Palliative and End-of-Life Care

Table 2 presents an overview of the palliative care and EOLC provided prior to death. In patients who died during Era 2, the rates of attempted cardiopulmonary resuscitation, intubation, and direct current shock at the end of life were significantly lower than in those who died in Era 1 (53% vs. 6%; *p* < 0.001, 47% vs. 0%; *p* < 0.001, 37% vs. 6%; *p* = 0.005, respectively). Compared to Era 1, a greater proportion of patients who died in Era 2 received opioids (11% vs. 70%). All the patients who received opioids were non-intubated and the administration was indicated to relieve dyspnea resistant to hemodynamic interventions (afterload reduction, diuretics, and inotropes). Sedative medications were significantly more commonly used in intubated patients who died in Era 1 and non-intubated patients who died in Era 2 (*p* < 0.05). Sedative medications without intubation were used for refractory symptoms, such as dyspnea, malaise, and delirium. Patients who died in Era 2 received more psychiatric consultations, although the difference was not significant. In Era 2, 70% of patients received multidisciplinary EOLC discussion support, a significant increase from 5% in Era 1 (*p* < 0.001).

### 3.3. Nutrition in the Three Days Prior to Death

Among the patients who died in Era 1 and Era 2, total parenteral nutrition was provided to 7 (37%) and 11 (33%), tube feeding to 1 (5%) and 5 (15%), and 9 (47%) and 12 (36%) patients fasted, respectively. There were no significant differences in nutrition administration methods between the two groups. Food intake three days before death was maintained in 9 out of 19 patients (47%) in Era 1 and 16 out of 33 patients (48%) in Era 2. The characteristics of the patients who maintained food intake three days prior to death are summarized in Table 3. Compared to the patients who died in Era 1 (n = 9), significantly more patients who died in Era 2 (n = 16) maintained food intake during opioid administration (11% vs. 63%, respectively, *p* = 0.013). In addition, food intake during sedation was not observed in Era 1 but was observed in 19% of patients who died in Era 2 (*p* = 0.166). Nutritional counseling was more frequently provided to patients who died in Era 2 than to those who died in Era 1 (33% vs. 81%, respectively, *p* = 0.017), and the change from a low-sodium diet to a regular-sodium diet was also significantly more frequent (59% vs. 22%, respectively, *p* = 0.025). Permission for non-hospital foods, such as bringing a patient’s favorite meal cooked by their family, also tended to be more frequently granted to patients who died in Era 2 as compared to those who died in Era 1 (*p* = 0.053).

Logistic regression analysis of factors associated with the administration of artificial nutrition three days prior to death was performed (Table 4). In the univariate regression analysis, the number of previous hospitalizations for HF (OR 0.69; 95% CI 0.51–0.93; *p* = 0.014) and HF due to NICM (OR 3.30; 95% CI 1.03–10.60; *p* = 0.045) were significantly associated with the administration of artificial nutrition in the final three days of life. Multivariate analysis demonstrated that the number of previous HF hospitalizations (OR 0.63; 95% CI 0.44–0.91; *p* = 0.014), NICM-caused HF (OR 15.8; 95% CI 2.42–103.00; *p* = 0.004), and multidisciplinary support for EOLC discussions (OR 0.15; 95% CI 0.03–0.91; *p* = 0.039) were independent factors related to the administration of artificial nutrition three days prior to death.

## 4. Discussion

In this study, we compared the changes in HF therapies in terms of palliative care and EOLC discussions before and after the initiation of HF-PCT activities at our institute, with a focus on food intake at the end of life. The major findings were that after HF-PCT activities, (a) fewer invasive procedures were performed, (b) the use of opioids and non-intubated sedatives for symptom relief increased, (c) support from the multidisciplinary team in EOLC discussions increased, and (d) quality of meals was improved at the end of life in patients (from low-sodium diet to regular diet, adequate symptom relief with opioids, provision of non-hospital meals such as patients’ favorite meals mainly by family members). Furthermore, the administration of artificial nutrition in the final three days prior to death was associated with NICM etiology, number of previous hospitalizations for HF, and multidisciplinary EOLC discussion support. To the best of our knowledge, this is the first report on changes in HF palliative care and end-of-life food intake after the initiation of HF-PCT activities.

### 4.1. End-of-Life Discussion with Patients with HF

Considering the plateau in diagnostic capacity and treatment efficacy for HF, new problems have arisen related to difficult EOLC decision-making under uncertain disease trajectories [14]. In addition, physicians often discuss EOLC with the families rather than the patients in Far East Asian countries such as China, South Korea, and Japan [15,16,17], where physicians and patients’ families traditionally tend to avoid giving unfavorable information to patients. However, Matsushima et al. showed that 85% of English-speaking Japanese Americans desired to make treatment decisions on their own, as compared to only 36% of Japanese individuals living in Japan [18]. Another Japanese study had shown that only 4.7% of patients with end-stage HF participated in EOLC discussions [19]. In the present study, EOLC discussions were more frequent in patients who died in Era 2. Moreover, the number of patients who underwent invasive procedures prior to death was significantly lower among patients who died in Era 2. Our findings suggest that HF-PCT activities might facilitate EOLC discussions based on patient values and preferences and could avoid unnecessary invasive treatment prior to death.

The current study did not confirm the existence of the ACP process. However, EOLC discussions with patients are an extension of the ACP process. The latest survey, conducted by the Japanese Ministry of Health, Labour and Welfare in 2017, indicated that 64.9% of Japanese individuals approved ACP and that 66% of them agreed to make an advance directive [20]. Furthermore, the newly revised Japanese Guidelines on the Diagnosis and Treatment of Acute and Chronic Heart Failure consider ACP as a class I recommendation for the management of HF [9]. Reflecting such situations, ACP and patient-centered decision-making processes have become an increased focus in Japan.

### 4.2. Symptom Management

In this study, many patients who died in Era 2 received opioids for refractory dyspnea. Kuragaichi et al. reported in a nationwide survey that dyspnea is the most common symptom requiring palliative care in patients with HF [10]. Some small studies have shown that low-dose opioids, especially morphine, relieve breathlessness in these patients [21,22], although another study does not support this finding [23]. Further studies are required to investigate appropriate opioid use in patients with HF and refractory dyspnea. Furthermore, there has been little evidence for palliative sedation in non-cancer patients, including HF [24]; however, in this study, 36% of non-intubated patients with HF who died in Era 2 received palliative sedation. Palliative sedation is only performed to relieve intractable distress at the end of life, but not to hasten death [25]. Clinicians should discuss the indication of palliative sedation with a multidisciplinary team such as an HF-PCT in terms of the patients’ benefits, goals, and risks, as well as the limited prognosis and presence of treatable factors. Psychological issues are also important problems in patients with HF. In particular, depression has an independent impact on morbidity and mortality in HF [26,27]. It is important to recognize psychological problems that may occur in the complex disease trajectory of HF. In this study, specialized psychiatric care was more often performed in patients with HF who died in Era 2; therefore, the HF-PCT may promote psychological support.

### 4.3. Diet in Palliative Care of Patients with HF

Food intake is extremely important in human life. Clinical evidence on the administration of artificial nutrition at the end of life is limited, even in oncology, and even more in HF. However, the goal of nutritional care at the end of life may be the same in both cancer and HF, in which it needs to change from maintaining nutritional status and function to ensure the patient’s well-being and quality of life [7]. In addition, the enjoyment of food may increase when restrictions are lifted. In Era 2 of this study, patients with HF, who had maintained food intake until three days prior to death, were provided symptomatic relief using opioids and dietary modification—providing a normal diet, not a low-sodium diet. Extensive communication and psychosocial support between the healthcare team and patients and/or families are important to alleviate distress related to food intake and weight loss, eliminate false expectations about nutrition, and set goals for nutritional care [28]. In this study, discussing EOLC with multidisciplinary support was associated with a decrease in the use of artificial nutrition in the last three days of life. If HF-PCT activities provide more opportunities to discuss EOLC in the future, the administration of unnecessary artificial nutrition may be avoided, and patients may be able to enjoy their meals at the end of life.

### 4.4. Limitations

The present study has several limitations. First, the retrospective and single-center nature of this study, including the small number of study patients, might have resulted in a certain extent of bias. Second, patients who die at a university hospital may be a selected group, incorporating bias. Thus, future studies should include a larger population and more hospitals. Third, although palliative care should be provided at the early stage of a life-threatening illness, only patients with HF who died at our hospital were included in this study. Further research is required to confirm whether palliative care would benefit patients from an earlier stage of HF. Fourth, since this study focuses on correlations between eras, reverse causality and hidden causal relationships may exist. Fifth, Era 1 and Era 2 differ not only in the existence of HF-PCT but also in the time background of HF palliative care awareness. The time background may be responsible for the differences found between the two eras. Sixth, because of the retrospective nature of the study, we could not evaluate objective health-related quality of life indicators in determining the effectiveness of HF-PCT consultation. Seventh, we will examine a longer period before death in the next study in near future. Finally, further research is required to examine how to provide HF palliative care that is adapted to the healthcare system in Japan.

## 5. Conclusions

Despite increasing attention on palliative care in HF, providing optimal palliative care at the end of life presents many challenges and complexities. The present study indicated that HF-PCT activities provide an opportunity to discuss EOLC with patients, reduce the burden of physical and mental symptoms, and may shift the goals of end-of-life nutritional care to ensuring comfort and quality of life.

## Figures and Tables

**Table 1 nutrients-13-02387-t001:** Baseline characteristics of patients.

	Total (n = 52)	Era 1 (n = 19)	Era 2 (n = 33)	*p* Value
Age, year	70.0 ± 15.1	69.5 ± 18.1	70.3 ± 13.4	0.861
Female, n (%)	22 (42)	8 (42)	14 (42)	0.982
Duration of HF, months	58.4 (10.9–173.2)	80.7 (2.0–207.8)	56.5 (11.8–158.4)	0.643
No. of previous HF hospitalization, n	3 (1–5)	3.0 (1.0–5.0)	3.0 (1.0–4.5)	0.992
Left ventricular ejection fraction, %	35 (20–56)	30 (16–52)	37 (20–63)	0.666
Intensive care unit hospitalization, n (%)	6 (12)	1 (5)	5 (15)	0.283
Etiology				
Ischemic, n (%)	7 (13)	4 (21)	3 (9)	0.223
Valvular, n (%)	12 (23)	5 (26)	7 (21)	0.674
Non-ischemic cardiomyopathy, n (%)	22 (42)	6 (32)	16 (48)	0.235
Pulmonary arterial hypertension, n (%)	8 (15)	1 (5)	7 (21)	0.125
Others, n (%)	3 (6)	3 (14)	0 (0)	0.019 *
Comorbidities				
Cerebrovascular disease, n (%)	9 (17)	5 (26)	4 (12)	0.193
Hypertension, n (%)	12 (23)	6 (32)	6 (18)	0.270
Diabetes, n (%)	21 (40)	7 (37)	14 (42)	0.693
Atrial fibrilation, n (%)	27 (52)	9 (47)	18 (55)	0.618
Malignancies, n (%)	5 (10)	1 (5)	4 (12)	0.419
Cardiac resynchronization therapy, n (%)	19 (37)	8 (42)	11 (33)	0.527
Implantable cardioverter defibrillator, n (%)	19 (37)	7 (37)	12 (36)	0.973
Systolic blood pressure, mmHg	104.0 ± 26.5	113.2 ± 29.0	98.7 ± 23.7	0.056
NYHA class III or IV, n (%)	43 (83)	15 (79)	28 (85)	0.592
Medication on admission				
ACE-I/ARB, n (%)	29 (56)	13 (68)	16 (48)	0.163
β-blocker, n (%)	31 (60)	11 (58)	20 (61)	0.848
Mineralocorticoid receptor antagonist, n (%)	25 (48)	7 (37)	18 (55)	0.219
Loop diuretic, n (%)	44 (85)	15 (79)	29 (88)	0.390
Laboratory data				
NT-proBNP, pg/mL	12,683 (5181–31,264)	12,684 (4512–38,571)	13,728 (5526–31,109)	0.770
Blood urea nitrogen, mg/dL	46.0 ± 24.9	52.1 ± 25.6	42.5 ± 24.2	0.180
Creatinine, mg/dL	1.8 ± 1.2	2.1 ± 1.4	1.6 ± 1.0	0.148
Sodium, mEq/L	135.0 ± 7.4	133.6 ± 8.1	135.8 ± 7.0	0.310

Data are presented as mean ± SD, median (IQR) and n (%); HF = heart failure; HFrEF = HF with reduced ejection fraction; NYHA = New York Heart Association functional classification; ACE-I = angiotensin-converting-enzyme inhibitor; ARB = angiotensin II receptor blocker; NT-proBNP = n-terminal b-type natriuretic peptide. * *p* < 0.05.

**Table 2 nutrients-13-02387-t002:** Palliative and end-of-life care.

	Era 1 (n = 19)	Era 2 (n = 33)	*p* Value
Length of hospital stay until death, days	20 (9–59)	24 (14–91)	0.448
Transfer to intensive care unit prior to death, n (%)	2 (11)	0 (0)	0.057
Invasive treatment at the time of death			
Cardiopulmonary resuscitation, n (%)	10 (53)	2 (6)	<0.001 *
Intubation, n (%)	9 (47)	0 (0)	<0.001 *
Direct current shocks, n (%)	7 (37)	2 (6)	0.005 *
Mechanical circulatory support, n (%)	1 (5)	0 (0)	0.183
Palliative care intervention			
Use of opioid, n (%)	2 (11)	23 (70)	<0.001 *
Use of sedative medication, n (%)	6 (32)	13 (39)	0.573
For intubated patients, n (%)	5 (26)	1 (3)	0.020 *
For non-intubated patients, n (%)	1 (5)	12 (36)	0.018 *
Specialized psychiatric care, n (%)	4 (21)	15 (45)	0.078
Multidisciplinary support for EOLC discussions, n (%)	1 (5)	23 (70)	<0.001 *

Data are presented as median (IQR) and n (%); EOLC = end-of-life care. * *p* < 0.05.

**Table 3 nutrients-13-02387-t003:** Characteristics of patients who maintained oral intake in the three days prior to death.

	Era 1 (n = 9)	Era 2 (n = 16)	*p* Value
Use of artificial nutrition, n (%)	0 (0)	3 (19)	0.166
Food intake under opioid administration, n (%)	1 (11)	10 (63)	0.013 *
Food intake under sedative medication administration, n (%)	0 (0)	3 (19)	0.166
Nutritional Counselling, n (%)	3 (33)	13 (81)	0.017 *
Change from low-sodium to regular-sodium diets, n (%)	2 (22)	11 (59)	0.025 *
Permission for non-hospital meals, n (%)	2 (22)	10 (63)	0.053
Use of oral nutritional supplements, n (%)	3 (33)	10 (63)	0.161

Data are presented as n (%). * *p* < 0.05.

**Table 4 nutrients-13-02387-t004:** Results of univariate and multivariate analysis associated with the administration of artificial nutrition three days prior to death.

	Univariate Analysis			Multivariate Analysis		
	OR	95% CI	*p* Value	OR	95% CI	*p* Value
Age	1.01	0.97–1.05	0.74	1.00	0.95–1.06	0.86
Female	1.20	0.39–3.70	0.76	1.22	0.27–5.58	0.80
ICM	0.60	0.11–3.44	0.57			
NICM	3.30	1.03–10.60	0.045 *	15.80	2.42–103.00	0.004 *
VHD	0.75	0.19–2.91	0.68			
NYHA class III or IV on admission	0.43	0.10–1.84	0.25			
Number of previous hospitalizations for HF	0.69	0.51–0.93	0.014 *	0.63	0.44–0.91	0.014 *
Multidisciplinary support for EOLC discussions	0.29	0.09–1.00	0.05	0.15	0.03–0.91	0.039 *

OR = odds ratio; CI = confidence interval; ICM = ischemic cardiomyopathy; NICM = non-ischemic cardio myopathy; VHD = valvular heart disease; HF = heart failure; EOLC = end-of-life care. * *p* < 0.05.

## Data Availability

Data supporting the results obtained in this study are available from the corresponding author upon reasonable request. All data were obtained from the subjects and are not available to the public for ethical reasons.

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
