# Peer review of "Multidisciplinary Team-Based Palliative Care for Heart Failure and Food Intake at the End of Life"

_nutrients, 2021, doi:10.3390/nu13072387_

Round 1

Reviewer 1 Report

This paper shows well the effect of palliative support (especially nutritional intervention) on morbidity in terminally ill heart failure patients (an ill-known feature in medicine altogether). I have some questions:

  • The last 3 days of life were analyzed for the use of artificial nutrition, but this seems quite short to me, and the authors should state more clearly that a longer examination period is warranted.
  • The number of patients with HF due to non-ischemic cardiomyopathy appears to be fairly high to me, especially in the ERA-2 group (although not significant). Is this a certain subpopulation?
  • Some of the numbers in this study are quite low. Has the overall statistical power been calculated?

Author Response

We thank Editor's specific comments and the Reviewer for her/his valuable comments. In line with the comments, we have revised our manuscript. Our detailed responses will follow the Editor and Reviewer’s comments. Our point-to-point responses are shown in the text in red to facilitate the review process.

Reviewer 1 comments:

This paper shows well the effect of palliative support (especially nutritional intervention) on morbidity in terminally ill heart failure patients (an ill-known feature in medicine altogether). I have some questions.

[Comment]

The last 3 days of life were analyzed for the use of artificial nutrition, but this seems quite short to me, and the authors should state more clearly that a longer examination period is warranted.

[Response]

Thank you very much for the valuable comment. We fully agree with the reviewer and we consider that a longer examination period is warranted; however, especially in this retrospective study, 3 days examination was appropriate to observe from acute exacerbation of end-stage heart failure to death, retrospectively. As the reviewer pointed out, we consider that 3 days might be short. We would examine longer period of time in the next study. We added this issue in the Limitation section.

Lines 278-279. Seventh, we will examine longer period before death in the next study in near future.

[Comment]

The number of patients with HF due to non-ischemic cardiomyopathy appears to be fairly high to me, especially in the ERA-2 group (although not significant). Is this a certain subpopulation?

[Response]

Thank you very much for the valuable comment. Probably because this study was performed in university hospital, we had some bias. We have already described this issue in the Limitation section (lines 265-267). We are planning a larger cohort study with some other hospitals, in which more ischemic cardiomyopathy will be included.

[Comment]

Some of the numbers in this study are quite low. Has the overall statistical power been calculated?

[Response]

Thank you very much for this comment. We have already indicated in the Limitation section. In the present study, we predicted that the percentage of EOLC discussions that received multidisciplinary support would be 20% in Era 1 and 70% in Era 2. Assuming an alpha error of 0.05, the required sample size was calculated to be 19, which might be fine in the present study.

Finally, we again would like to thank the Reviewer for the valuable comments on our work. We sincerely hope that our revised manuscript may again be considered for publication in the Journal.

Reviewer 2 Report

The topic of a palliative care for heart failure is novel and the article is original. It can be published after a minor revision. There are few suggestions:

INTRODUCTION:

- line 75: the letter “e” is missing in the word “life”, please correct.

MATERIALS AND METHODS:

- a report of ethical approval and informed consent is missing.

- generally this section should be described in a more detailed way; e. g. which nutritional parameters had been studied (line 99), etc.

RESULTS:

  • line 129 (and Table 1): why the NT – proBNP is exprressed as median and not as mean (as the other laboratory data)? Please clarify.
  • line 142: please add the p - value.
  • lines 148 - 149: the information about the use of sedative medications shoulb be reported in methods.
  • lines 150 - 152: please clarify from which value is the increase and add the p - value.
  • lines 156 - 158: it is not clear how many patients mantained oral intake in the 3 days prior to death, please clarify.

DISCUSSION:

  • line 193: please clarify how did you demonstrate the improved quality meals in era2?
  • lines 213 - 217: the reference is missing.

Author Response

We thank Editor's specific comments and the Reviewer for her/his valuable comments. In line with the comments, we have revised our manuscript. Our detailed responses will follow the Editor and Reviewer’s comments. Our point-to-point responses are shown in the text in red to facilitate the review process.

Reviewer 2 comments:

The topic of a palliative care for heart failure is novel and the article is original. It can be published after a minor revision. There are few suggestions:

[Comment]

INTRODUCTION:

- line 75: the letter “e” is missing in the word “life”, please correct.

[Response]

We apologize for the typo. We have corrected this issue.

[Comment]

MATERIALS AND METHODS:

- a report of ethical approval and informed consent is missing.

[Response]

Thank you for this comment. Ethical approval and informed consent have been described.

Lines 292-296: Institutional Review Board Statement: The study was conducted in accordance with the guidelines of the Declaration of Helsinki and approved by the Institutional Review Board of Kurume University (No. 18066).

Informed Consent Statement: The release of research information in this study allowed us to collect and analyze data without obtaining written informed consent from each patient.

[Comment]

- generally this section should be described in a more detailed way; e. g. which nutritional parameters had been studied (line 99), etc.

[Response]

We apologize for this issue. The nutritional parameters were not mentioned in the text. We removed “nutritional parameters” and changed the description of the Method section in more detail.

Lines 92-95: The HF-PCT was available for all HF patients and assisted the treatment of patients with HF through refractory symptom relief, establishment of care goals, psychosocial support, support of ACP and EOLC discussions, and provision of nutritional support for EOLC.

Lines 97-110: We obtained data on the patients’ background, etiology of HF, duration of HF, comorbidities, echocardiographic findings, HF treatments, sedative medications, laboratory data, location of death, invasive procedures undergone before death (cardiopulmonary resuscitation, intubation, direct current shocks and Mechanical circulatory support) and length of hospital stay. We also assessed differences in the palliative care intervention, such as the use of opioids and sedative medications for refractory symptoms, psychiatric support, and multidisciplinary support for EOLC discussions, between the patients who died in Era 1 and those who died in Era 2. To assess the nutritional status of patients with end-stage HF, we examined the method of receiving nutrition 3 days prior to death and nutritional interventions (discontinuation of salt reduction, allowing non-hospital meals, and use of oral nutritional supplements) for patients who were maintaining oral intake at the time. We also examined factors associated with the use of artificial nutrition (total parenteral nutrition and tube feeding) 3 days prior to death.

[Comment]

RESULTS:

line 129 (and Table 1): why the NT – proBNP is exprressed as median and not as mean (as the other laboratory data)? Please clarify.

[Response]

Thank you for this comment. Because NT-proBNP was not normally distributed, we have shown the median value.

[Comment]

line 142: please add the p - value.

[Response]

Thank you for this comment. We added the P value.

Lines 146-147: who died in Era 1 (53% vs. 6%; p<0.001, 47% vs. 0%; p<0.001, 37% vs. 6%; p=0.005, respectively).

[Comment]

lines 148 - 149: the information about the use of sedative medications should be reported in methods.

[Response]

Thank you for this comment. According to the Reviewer’s suggestion, we added this issue in the Methods section.

Lines 97-100: We obtained data on the patients’ background, etiology of HF, duration of HF, comorbidities, echocardiographic findings, HF treatments, sedative medications, laboratory data, location of death, invasive procedures undergone before death

[Comment]

lines 150 - 152: please clarify from which value is the increase and add the p - value.

[Response]

Thank you for this comment. We have revised this issue.

Lines 153-156: Patients who died in Era 2 received more psychiatric consultations, although the difference was not significant. In Era 2, 70% of patients received multidisciplinary EOLC discussion support, a significant increase from 5 percent in Era 1 (p<0.001).

[Comment]

lines 156 - 158: it is not clear how many patients maintained oral intake in the 3 days prior to death, please clarify.

[Response]

Thank you for this valuable comment. According to the Reviewer’s suggestion, we have revised this issue.

Lines 163-164: Food intake 3 days before death was maintained in 9 out of 19 patients (47%) in Era 1 and 16 out of 33 patients (48%) in Era 2.

[Comment]

DISCUSSION:

line 193: please clarify how did you demonstrate the improved quality meals in era2?

[Response]

Thank you for this valuable comment. In patients with heart failure are usually forced to eat low-sodium diet; however, in Era 2, patients were able to eat regular-sodium diet or non-hospital meals (their favorite meals, mainly from families), which we considered “the improved quality meals”.

According to this comment, we have revised this issue in lines 197-200: (d) quality of meals was improved at the end of life in patients (from low-sodium diet to regular diet, adequate symptom relief with opioids, provision of non-hospital meals such as patients' favorite meals mainly by family members).

[Comment]

lines 213 - 217: the reference is missing.

[Reference]

We apologize for this typo. We have added ref. 20, the URL of the Ministry of Health, Labour and Welfare's web page.

Finally, we again would like to thank the Reviewer for the valuable comments on our work. We sincerely hope that our revised manuscript may again be considered for publication in the Journal.

Reviewer 3 Report

The topic is of interest, the study is well conducted and the results are clearly presented.

My question to the authors is:

Were patients on LVAD support among study population?

Author Response

We thank Editor's specific comments and the Reviewer for her/his valuable comments. In line with the comments, we have revised our manuscript. Our detailed responses will follow the Editor and Reviewer’s comments. Our point-to-point responses are shown in the text in red to facilitate the review process.

Reviewer 3 comment:

The topic is of interest, the study is well conducted and the results are clearly presented.

[Comment]

Were patients on LVAD support among study population?

[Response]

Thank you for this important comment. We included no patient on LVAD, because none of patients with LVAD during the study period died in our hospital.

Finally, we again would like to thank the Reviewer for the valuable comments on our work. We sincerely hope that our revised manuscript may again be considered for publication in the Journal.